# Impact of the Pre-Transplant Circulatory Supportive Strategy on Post-Transplant Outcome: Double Bridge May Work

**DOI:** 10.3390/jcm10204697

**Published:** 2021-10-13

**Authors:** Nai-Kuan Chou, Heng-Wen Chou, Chuan-I Tsao, Chih-Hsien Wang, Kevin Po-Hsun Chen, Nai-Hsin Chi, Shu-Chien Huang, Hsi-Yu Yu, Yih-Sharng Chen

**Affiliations:** 1Division of Cardiovascular Surgery, Department of Surgery, National Taiwan University Hospital, and College of Medicine, National Taiwan University, Taipei 100, Taiwan; nick.chounaikuan@gmail.com (N.-K.C.); ntuhcvs@gmail.com (H.-W.C.); wchemail@gmail.com (C.-H.W.); chinaihsin@gmail.com (N.-H.C.); dtsurg99@yahoo.com.tw (S.-C.H.); hsiyuyu@gmail.com (H.-Y.Y.); 2Graduate Institute of Clinical Medicine, College of Medicine, National Taiwan University, Taipei 100, Taiwan; 3Department of Nursing, National Taiwan University Hospital, Taipei 100, Taiwan; chuani930@gmail.com; 4School of Medicine, Auckland University, Auckland 1023, New Zealand; a70541020@gmail.com; 5College of Medicine, National Taiwan University, Taipei 100, Taiwan; 6Cardiovascular Center, National Taiwan University Hospital, Taipei 100, Taiwan

**Keywords:** mechanical circulatory support, heart transplantation, survival curve, ventricular assist device, extracorporeal membrane oxygenation, timing

## Abstract

Background: The number of waitlisted patients requiring mechanical circulatory support (MCS) as a bridge to heart transplantation is increasing. The data concerning the results of the double-bridge strategy are limited. We sought to investigate the post-transplant outcomes across the different bridge strategies. Methods: We retrospectively reviewed a heart transplantation database from Jan 2009 to Jan 2019. Intra-aortic balloon pump (IABP), extracorporeal membrane oxygenation (ECMO), and ventricular assist devices (VAD) were the MCS that we investigated. The pre- and post-transplant characteristics and variables of patients bridged with the different types of MCS were collected. The post-transplant survival was compared using Kaplan–Meier survival analysis. Results: A total of 251 heart transplants were reviewed; 115 without MCS and 136 with MCS. The patients were divided to five groups: Group 1 (no MCS): *n* = 115; Group 2 (IABP): *n* = 15; Group 3 (ECMO): *n* = 33; Group 4 (ECMO-VAD): double-bridge (*n* = 59); Group 5 (VAD): *n* = 29. Survival analysis demonstrated that the 3-year post-transplant survival rates were significantly different among the groups (Log-rank *p* < 0.001). There was no difference in survival between group 4(ECMO-VAD) and group 1(no MCS)1 (*p* = 0.136), or between group 4(ECMO-VAD) and group 5(VAD) (*p* = 0.994). Group 3(ECMO) had significantly inferior 3-year survival than group 4(ECMO-VAD) and group 5(VAD). Conclusion: Double bridge may not lead to worse mid-term results in patients who could receive a transplantation. Initial stabilization with ECMO for critical patients before implantation of VAD might be considered as a strategy for obtaining an optimal post-transplant outcome.

## 1. Introduction

Heart transplantation remains the gold standard for patients with end-stage heart failure (HF) and a poor response to optimal medical therapy [1]. However, only a limited number of patients on the waiting list receive this procedure annually due to the shortage of the organ [2]. Mechanical circulatory support (MCS) is mandatory to bridge the patients with decompensated heart failure to heart transplantation, and owing to the advancement in MCS, waiting list mortality has declined in recent years [2].

Ventricular assist devices (VADs) have been proven to improve survival in advanced HF and is effective as a bridge to heart transplantation, with favorable life quality and adverse events [3,4]. Intra-aortic balloon pump (IABP) offers partial circulatory support for advanced heart failure and could be a bridge to VAD implantation or transplantation in some studies [5,6]. Extracorporeal membrane oxygenation (ECMO) has been widely used as the first-line rescue for cardiogenic shock (CS) and has demonstrated better survival in cardiac arrest than conventional cardiopulmonary resuscitation (CPR) [7]. ECMO offers the advantages of rapid-implantation, and it is less expensive than VAD and can also be used as a bridge to decision, to candidacy, or to transplantation [8].

Based on the availability of devices and donated organs, physicians may follow different strategies of MCS for different clinical scenarios [8]. The difference in survival could be largely attributed to pre-MCS condition and Interagency Registry for Mechanically Assisted Circulatory Support (INTERMACS) Profiles [9].

The double bridge strategy is increasingly being adopted. For IMTERMACS profile 1 and CS patients, initial ECMO therapy with crossover to VAD is associated with improved outcomes [10,11,12]. The double bridge strategy in heart transplantation can provide a window or screening tool to select a more suitable condition of recipients to receive a donor heart in a short-term outcome that might be helpful for transplant center program results. In the present study, we investigate the longer outcome and survival associated with the double bridge strategy and different MCS statuses. In this study, we aim to explore the post-transplant survival of different MCS strategies.

## 2. Materials and Methods

### 2.1. Data Collection and Study Population

Data were prospectively collected and retrospectively reviewed from our database of heart transplantation between Jan 2009 and Jan 2019, which was approved by our institutional review board (NTUH 201510022 RIND).

The MCS included IABP, ECMO, temporary VAD (tVAD), and durable VAD (dVAD). We categorized patients into 5 groups based on the MCS they received before transplants: Group 1 (no MCS): patients without any MCS; Group 2 (IABP only): patients were supported with IABP only; Group 3 (ECMO): patients were supported with ECMO to bridge to heart transplantation; Group 4 (ECMO-VAD): patients were supported with ECMO initially and then converted to VAD; and Group 5 (VAD): patients were implanted with VAD initially. Patients who initially received IABP and were then shifted to ECMO or VAD for adequate circulatory support were categorized into Group 3 (ECMO) or Group 5 (VAD), respectively.

### 2.2. Selection of MCS and Procedures

The selection of MCS devices was largely dependent on the hemodynamic condition and reserve of heart function in patients with end-stage HF and the waitlisted duration for heart transplant.

The selection of the types of MCS was a process of decision making that included the status of numbers of organs dysfunction, the status of the INTEERMAC, the availability of the MCS, the supporting capacity of the different MCS, and the predicted waiting time. Although there were lots of uncertain factors remaining to consider, we will attempt to describe the general rule that we usually applied.

IABP was the first MCS that we took into consideration. However, because of the limited additional supportive capacity, it was reserved for patients with a mild hemodynamic disturbance under a high dose of inotropes. After that, ECMO or VAD were implanted if the circulatory support of IABP failed to achieve an adequate hemodynamic status. In general, we applied inotropic equivalent (IE, µg/kg/min. = dopamine + doubutamine + epinephrine × 100 + norepinephrine × 100) as a parameter; when IE over 15 and systolic pressure < 90 mmHg, IABP was considered. When persistent end-organ dysfunction after IABP or IE > 20, further advanced MCS were considered.

For patients of INTERMACS profile 1 with CS or under CPR, ECMO was set up as the resuscitation of choice. The ECMO equipment, management, and exclusion criteria have been described in previous studies [13]. If a lung edema or marked LV distension developed without aortic valve opening despite optimal medication after ECMO, a LVAD was required. In this double-bridge strategy, we usually chose a tVAD (Levitronix CentriMag^®^) as a bridge to recovery, decision or transplantation. For patients without signs of heart function recovery and for whom heart transplantation was inevitable, we also shifted to VAD after stabilization of renal and liver function.

For patients of INTERMACS profile 2 or 3 who were inotrope-dependent, we applied VAD as a bridge to heart transplantation. The VAD systems that we adopted included temporary VAD (tVAD): CentriMag™ (Abbott, Santa Clara, Calif) and durable VAD (dVAD): HeartMate II™, HeartMate 3™ (Abbott Laboratories, Lake Bluff, Ill) and HeartWare™ (Medtronic, Mounds View, MN, USA). We usually placed the outflow cannulae at the ascending aorta or subclavian artery, with inflow from the left ventricular apex for tVAD. The implanting procedure for dVAD was performed as recommended [14].

The protocol for heart transplantation in our institute included ABO blood-type matching and a pre-transplant in vivo crossmatch test. Since the MCS was sometimes associated with allosensitization [15], we applied desensitization therapy with preoperative and intraoperative plasma exchange, immunoglobulin (IVIG) infusion, and/or anti-CD 20 antibody administration for patients with repeated positive crossmatch.

### 2.3. Follow-Up Protocol

Endomyocardial biopsy (EMB) was performed weekly during the first month after transplantation, then quarterly for 12 months, and yearly thereafter. All patients were followed monthly at a special cardiac transplantation clinic.

### 2.4. Statistical Analysis

Continuous variables are presented as means ± standard deviation and were compared using either Student’s *t*-testing or the Mann–Whitney U testing, as appropriate. Categorical variables are presented as proportions and were compared using Fisher’s exact testing or Pearson’s chi-square test. ANOVA was used to examine differences among the five groups. The Kaplan–Meier method with the log-rank test was used to estimate and compare 3-years survival, with time zero as the date of heart transplantation and patients censored at the time of death. Pairwise log-rank tests were used for multiple comparisons. The *p* values < 0.05 were considered to be statistically significant for all comparisons. Statistical analyses were performed with SPSS 19 (SPSS, Inc., Chicago, IL, USA).

## 3. Results

### 3.1. Characteristics and Post-Transplant Outcomes of MCS and Non-MCS Patients

There was a total of 251 heart transplants in our center. Among them, 136 patients received MCS bridging to heart transplantation, whereas 115 patients did not. Patients’ characteristics and donor status are shown in Table 1. Patients with MCS had a significantly more critical UNOS (the United Network for Organ Sharing) status, higher incidence of CPR, and shorter listing duration than those without MCS.

The transplant procedures and post-transplant outcomes are shown in Table 2. Compared to patients without MCS, the MCS group had inferior hospital, 90-day, 1-year post-transplant survival (Table 2). There was no significant difference in episodes of acute cellar rejection (ACR ≥ 1B) [16] or antibody-mediated rejection (AMR) between the two groups (Table 2). The total bilirubin and creatinine levels were not significantly different between the two groups after heart transplantation (Appendix A).

### 3.2. Characteristics and Outcomes of Different Types of MCS

Based on the types of MCS, the patients were divided into five groups: Group 1 (no MCS): 115 patients; Group 2 (IABP only): 15 patients; Group 3 (ECMO): 33 patients; Group 4 (ECMO-VAD): 59 patients, including tVAD 54, dVAD 5; Group 5 (VAD): 29 patients, including tVAD 23, and dVAD 5. With advances in MCS and the increased availability of machines, our preference of MCS changed from IABP and ECMO to VAD and the double-bridge strategy in recent years (Figure 1). The mean days from MCS implant to heart transplantation is shown in Figure 2.

The demographic data and pre-transplant status for each MCS group are shown in Table 3. Smoking and previous cardiac surgery were significantly different among the subgroups. The higher frequency of CPR in group 3 and group 4 could be mainly attributed to the implementation of ECPR. The waiting periods (wait from index admission, listing duration, and wait from MCS) were also significantly different between the groups. Group 5 (VAD) had the longest waiting period, especially for patients with durable VAD (Table 3 and Figure 2). In group 4, the days of conversion from ECMO to VAD was 12 ± 12 days. The incidence of pre-transplant hemodialysis was also significantly different between the groups, with a higher incidence in group 3 (ECMO) and group 4 (ECMO-VAD) (Table 3). The pre-transplant creatinine levels were also higher in these two groups (group 3, group 4) than that in the other groups, although the difference between groups was not statistically significant (*p* = 0.251) (refer to Appendix A).

The index procedures and outcomes after heart transplantation for each MCS group are shown in Table 4. Desensitization therapy was more frequently required in group 4 and group 5, while no patients in group 3 received desensitization before transplant.

However, when we focused on the patients in the VAD-supported (group 4 and group 5, total 88) group and non-VAD supported groups (group 2 and group 3, total 48), the VAD-supported group had significantly better hospital survival than the non-VAD group (*p* = 0.005). The analysis reconfirmed that the recipient supported with VAD had a better short-term outcome than those supported with IABP and ECMO only, although the basic demographic data had difference in the factors of previous cardiac surgery and CPR (Table 3).

There were no significant differences among the groups in terms of episodes of rejection, including a cellular rejection grade greater than 1B and AMR (Table 4), or in bilirubin and creatinine levels (Appendix A).

### 3.3. Kaplan–Meier Survival Analysis for 3 Years Follow-Up

The Kaplan–Meier estimates for patients’ survival within three years after heart transplant are displayed in Figure 3. Patients without pre-transplant MCS had significantly better 3-years survival than those with MCS (Figure 3A). Figure 3B shows Kaplan–Meier survival over three years after transplant for patients supported with different MCS devices (Log-rank *p* < 0.01). In the log-rank pairwise comparison of survival at three years, ECMO-bridged patients had lower survival than those without MCS (*p* < 0.001), those bridged with VAD (*p* = 0.037), and those with a double-bridge (ECMO-VAD) (*p* = 0.007). There was no significant survival difference between the double-bridge and VAD group (*p* = 0.994) or between double-bridge patients and those without MCS (*p* = 0.136).

## 4. Discussion

In this study, we assessed the impact of different types of MCS and bridge strategies on post-transplant outcomes. Patients who required MCS as a bridge to transplantation had inferior post-transplant survival compared to those who did not. Among the different types of MCS, ECMO as the direct bridge had the worst post-transplant survival, similar to the results of previous studies [8,17]. The outcomes for the ECMO bridge could be largely attributed to the more critical condition of patients before ECMO implant.

Our data also revealed that the post-transplant outcomes of tVAD were not inferior to those of dVAD, which may be related to waiting list and allocation policy. In addition, most of the VADs implanted were tVAD rather than dVAD, especially for the double-bridge group because of our insurance and cost issues. The patients with tVAD also had fewer waitlisted days than those with dVAD (Table 3). In general, the mean waiting duration for UNOS 1A in our area is 31.63 ± 36.27 days.

The CPR cases were more challenging in those that may be considered as heart recipients. In general, ECMO was initialized at the episode of CPR, then converted to VAD or to directly wait for a heart when neurological status was acceptable after further assessment. There were 10 patients in group 5 (VAD only) that experienced CPR, but they fortunately returned to spontaneous circulation without requiring ECMO rescue, and therefore could be implanted with VAD emergently. However, due to economic issues, not everyone can afford to apply dVAD (tVAD 23, and dVAD 5). In addition, we did not have any patients bridging from tVAD to dVAD, because the costs of the only tVAD were reimbursed by the insurance.

Multivariate analysis was performed regarding the pretransplant factors; the duration between MCS and HTx (days) and the pre-transplant CPR were the significant risk factor for hospital survival (*p* = 0.04 and 0.003, individually). The MCS strategy did not reach statistically significant difference (*p* = 0.058). Therefore, we believe that it was the duration of MCS that led to differences in hospital survival, and ECMO was the short-term device for MCS.

We also reviewed the ECMO database that was considered for listing heart transplants (Figure 4). Only 33.1% of them survived until discharge and less than 20% ultimately received the transplant. It is also noteworthy that only 19.7% of patients had the opportunity to receive a transplant among the ECMO-supported patients considered for the waitlist. This also demonstrates that there is always selection bias in the setting of these kinds of study.

The double-bridge concept has gradually developed in recent years, including in our hospital (Figure 1). Cheng [10] advocated for early crossover to VAD to improve survival and hospital discharge. The days of crossover in our hospital was 12 ± 12 days (Figure 2). Pre-operative stabilization with ECMO explained the similar post-transplant survival between the double-bridge and VAD groups, although a higher percentage of the double-bridge group had pre-transplant CPR and renal replacement therapy.

According to our findings, advanced MCS or double bridge might not increase risk of HTx (Figure 3). Conversely, the percentage of survival may be not inferior to that of patients supported with IABP. We also clearly described the duration of the different supported MCS to HTx and the duration in bridge. This provides a reference time point for clinicians to make decision in conversion from some MCS to more advanced ones.

Finally, in clinical practice, we will find any MCS to offer adequate support where possible, and if the support is inadequate, the further advanced MCS will be applied as soon as possible. We tried to demonstrate the principle of adequate support for the end organs function to achieve the criteria of HTx.

## 5. Limitation

This study is a single-center, retrospective record review with limited case numbers, and the results must be interpreted with caution. In addition, the choice of MCS and the timing of conversion depended on the surgeons’ preference, despite there being general principles to follow.

In fact, 36% (35 out of 97) of patients died while on the waiting list in the double-bridge group. The waitlisted survival for double bridge seemed to not be as optimistic as that for the ECMO group (41% died on ECMO) (Figure 4).

## 6. Conclusions

Patients with pre-transplant MCS have poorer post-transplant outcomes than those without MCS. Different strategies of MCS might convey different post-transplant survival. The VAD group, with either tVAD or dVAD, undoubtedly had the best post-transplant outcomes, while the ECMO groups has the worst outcomes among the MCS group. The double-bridge strategy, which is a means to stabilize patients of INTERMACS profile 1 or hemodynamic collapse with ECMO before switching to VAD, offered a post-transplant mid-term survival that was not inferior to that of the VAD group.

## Figures and Tables

**Figure 1 jcm-10-04697-f001:**
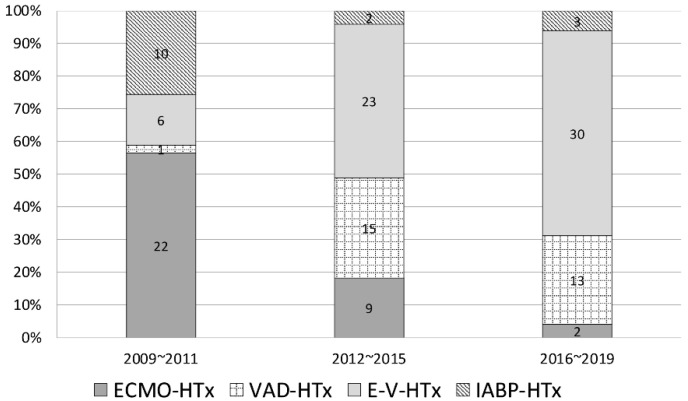
The distribution of every type of mechanical circulatory support in different period. ECMO: extracorporeal membrane oxygenation; HTx: heart transplantation; VAD: ventricular assist device; E-V-HTx: ECMO-VAD-heart transplant; IABP: intra-aortic balloon pump.

**Figure 2 jcm-10-04697-f002:**
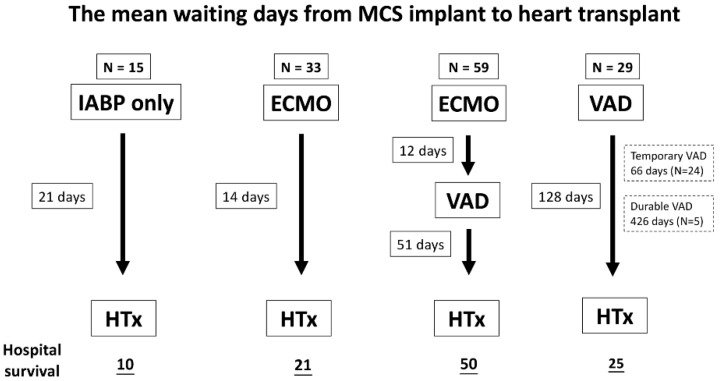
The mean waiting days from implantation of mechanical circulatory support devices to heart transplant. MCS: mechanical circulatory support; ECMO: extracorporeal membrane oxygenation; HTx: heart transplantation; IABP: intra-aortic balloon pump; VAD: ventricular assist device.

**Figure 3 jcm-10-04697-f003:**
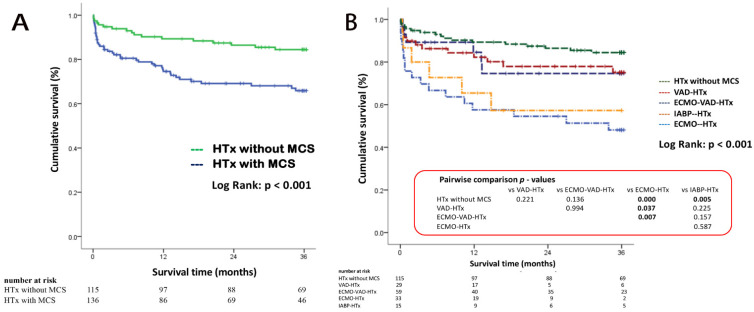
Kaplan–Meier estimate of post-transplant survival within 3 years between patients with and without MCS (**A**), and stratified by device types (**B**). Pairwise survival was compared usin log-rank tests. HTx: heart transplantation; MCS: mechanical circulatory support; ECMO: extracorporeal membrane oxygenation; VAD: ventricular assist device; IABP: intra-aortic balloon pump.

**Figure 4 jcm-10-04697-f004:**
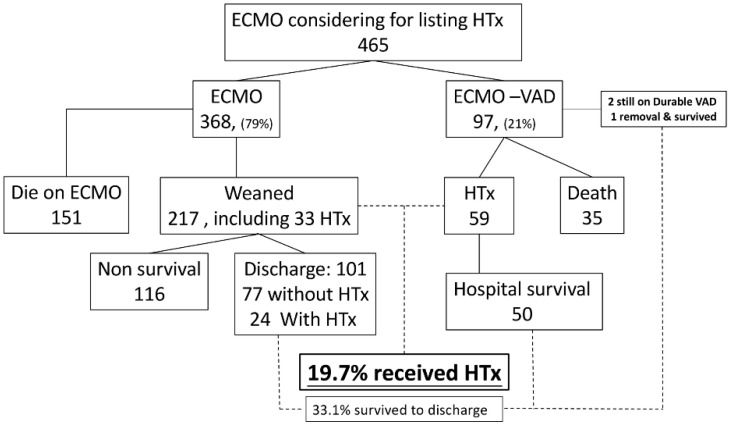
Outcome for ECMO patients considered to be listed as a heart recipient. HTx: heart transplantation; ECMO: extracorporeal membrane oxygenation; VAD: ventricular assist device.

**Table 1 jcm-10-04697-t001:** Patients’ pretransplant basic demographic data and donor data.

	Without MCS	With MCS	*p*	Total
Patients, *n* (%)	115 (45.8)	136 (54.2)		251
Gender, male *n* (%)	95 (82.6)	116 (85.3)	0.342	211 (84.1)
Age, years (mean ± SD)	44.1 ± 18.3	47.4 ± 14.3	0.113	45.9 ± 16.3
Blood type			0.006	
Type O, *n* (%)	23 (20)	52 (38.2)		75 (29.9)
Non-type O, *n* (%)	92 (80)	84 (61.8)		176 (70.1)
Hypertension, *n* (%)	28 (24.3)	44 (32.4)	0.104	72 (28.7)
Diabetes mellitus, *n* (%)	27 (23.5)	37 (27.2)	0.299	64 (25.5)
Hyperlipidemia, *n* (%)	19 (16.5)	30 (22.1)	0.173	49 (19.5)
Smoking, *n* (%)	27 (23.5%)	41 (30.1)	0.149	68 (27.1)
Primary etiology			0.064	
ICMP, *n* (%)	39 (33.9)	60 (44.1)		99 (39.4)
Non-ICMP, *n* (%)	76 (66.1)	76 (55.9)		152 (60.6)
Previous cardiac surgery, *n* (%)	40 (34.8)	49 (36.0)	0.471	89 (35.5)
Re-Tx, *n* (%)	3 (2.6)	5 (3.7)	0.457	8 (3.2)
UNOS status			0.000	
1A, *n* (%)	4 (3.5)	136 (100)		140 (55.6)
1B, *n* (%)	45 (39.1)	0 (0)		45 (17.9)
CPR *, *n* (%)	6 (5.2%)	73 (53.2%)	0.000	79 (32.2%)
Listing duration ^#^ Days, median(IQR)	253 (67~546)	54 (16–134)	0.005	89 (25–366)
Donor data				
Gender, male *n* (%)	84 (73)	94 (69.1)	0.495	178 (70.9)
Age, years (mean ± SD)	34.9 ± 15.0	37.7 ± 12.8	0.113	36.4 ± 13.9
Blood type			0.000	
Type O, *n* (%)	29 (25.2)	81 (59.6)		100 (43.8)
Non-type O, *n* (%)	86 (74.8)	55 (40.4)		141 (56.2)
Etiologies in brain death			0.064	
Head trauma, *n* (%)	51 (44.3)	66(48.5)		117(46.6)
Cerebrovascular/stroke, *n* (%)	38 (33.0)	46 (33.8)		84 (33.5)
Brain tumor, *n* (%)	2 (1.7)	4 (2.9)		6 (2.4)
Hypoxia, *n* (%)	16 (13.9)	14 (10.3)		30 (12.0)
Other, *n* (%)	8 (7.0)	6 (4.4)		14 (5.6)
CPR before donation, *n* (%)	34 (29.6)	38 (27.9)	0.952	72 (28.5)
Hypotension episode, *n* (%)	42 (36.5)	54 (39.7)	0.862	96 (37.9)
LVEF, % (mean ± SD)	65.3 ± 9.3	64.4 ± 10.1	0.434	64.8 ± 9.7

CPR *: cardiopulmonary resuscitation, ^#^: the waiting duration between listing and transplantation; CVA: cerebral vascular accident; DM: diabetes mellitus; ICMP: ischemic cardiomyopathy; LVEF: left ventricular ejection fraction; MCS: mechanical circulatory support; Re-Tx: re-transplantation; SD: standard deviation; UNOS: United Network for Organ Sharing; Listing duration ^#^: duration between listing and being transplanted, median, (IQR: interquartile rang, 25 percentile–75 percentile).

**Table 2 jcm-10-04697-t002:** Index transplantation procedure data in the study group.

	Without MCS Group 1	With MCS Group 2–5	*p*	Total
Patients, *n* (%)	115 (45.8)	136 (54.2)		251
H/D before HTx, *n* (%)	9 (8)	66 (49.3)	0.000	75 (30.4)
Ischemic time, min (mean ± SD)	148.1 ± 63.3	175.0 ± 68.6	0.002	163 ± 67
CPB duration, min (mean ± SD)	139.7 ± 61.0	172.5 ± 68.9	0.000	158 ± 67
Procedure duration, min (mean ± SD)	315.3 ± 125.4	345.7 ± 116.2	0.050	332 ± 121
Desensitization in HTx, *n* (%)	9 (7.8)	18 (13.2)	0.120	27 (10.6)
Post-HTx MCS, *n* (%)	21 (18.3)	50 (36.8)	0.001	71 (28.3)
Requiring H/D after HTx, *n* (%)	24 (21.2)	65 (48.5)	0.000	89 (36.0)
Extubation after HTx, day (mean ± SD)	4.6 ± 6.4	7.8 ± 9.9	0.023	6.2 ± 8.5
ICU stay after HTx, day (mean ± SD)	13.2 ± 8.3	18.9 ± 15.5	0.002	16.1 ± 12.9
Hospital survival, *n* (%)	106 (92.2)	106 (77.9)	0.001	212 (84.5)
90-day survival (%)	94.8	83.7	0.004	88.8
1-year survival (%)	89.3	74.6	0.002	81.4
3-year survival (%)	84.4	65.9	0.002	74.1
ACR ≥ 1B in 3 years, *n* (%)	8 (7.5)	12 (9.8)	0.356	20 (8.8)
AMR in 3 years, *n* (%)	7 (6.5)	5 (4.1)	0.297	12 (5.2)

ACR: acute cellular rejection; AMR: antibody-mediated rejection; H/D: hemodialysis; HTx: heart transplantation; CPB: cardiopulmonary bypass; MCS: mechanical circulatory support; SD: standard deviation.

**Table 3 jcm-10-04697-t003:** Demographic data of MCS bridge to HTx subgroups.

	IABP Group 2	ECMO Group 3	ECMO-VAD Group 4	VAD Group 5	*p*	Total
Patient, *n* (%)	15 (11)	33 (24.3)	59 (43.4)	29 (21.3)		136
Male, *n* (%)	15 (100.0)	28 (84.8)	47 (79.7)	26 (89.7)	0.210	116 (85.3)
Age, (years ± SD)	54.0 ± 9.9	47.5 ± 16.5	44.2 ± 14.5	50.4 ± 11.8	0.058	47.4 ± 14.3
Blood type					0.571	
Type O, *n* (%)	4 (26.7)	14 (42.4)	24 (40.7)	10 (34.5)		52 (38.2)
Non type O, *n* (%)	11 (73.3)	19 (57.6)	35 (59.3)	19 (65.5)		74 (61.8)
Hypertension, *n* (%)		13 (39.4)	20 (33.9)	8 (27.6)	0.540	44 (32.4)
DM, *n* (%)	5 (33.3)	12 (36.4)	11 (18.6)	9 (31.0)	0.253	37 (27.2)
Hyperlipidemia, *n* (%)	3 (20.0)	8 (24.2)	10 (16.9)	9 (31.0)	0.497	30 (22.1)
Smoking, *n* (%)	4 (26.7)	4 (12.1)	20 (33.9)	13 (44.8)	0.036	41 (30.1)
Primary etiology					0.638	
ICMP, *n* (%)	8 (53.3)	15 (45.5)	27(45.8)	10 (34.5)		60 (44.1)
Non ICMP, *n* (%)	7 (46.7)	18 (54.5)	32 (54.2)	19 (65.)		76 (55.9)
Previous cardiac surgery, *n* (%)	3 (20.0)	14 (42.4)	16 (27.1)	16 (55.2)	0.031	49 (36)
CPR *, *n* (%)	6 (42.9)	21 (67.7)	36 (63.2)	10 (35.7)	0.035	73 (56.2)
Waiting days from index admission to HTx, days (days ± SD)	42.1 ± 47.9	19.7 ± 18.0	52.8 ± 47.2	127.9 ± 311.6	0.000	47.2 ± 47.7
Listing duration ^#^, days, median (IQR)	59 (14–156)	16 (11–295)	53 (18–81)	112 (45–173)	0.017	54 (16–134)
Waiting days from MCS to HTx, day (mean ± SD)	21 ± 15	14 ± 9	63 ± 56	128 ± 312	0.019	60 ± 152
			ECMO-VAD 12 ± 12 VAD-HTx 51 ± 53	Temporary VAD 66 ± 78 Durable VAD 426 ± 714		
H/D before HTx, *n* (%)	2 (14.3)	16 (48.5)	39 (67.2)	9 (31.0)	0.000	66 (49.3)

CPR *: cardiopulmonary resuscitation, cardiac massage during waiting before transplantation; ECMO: extracorporeal membrane oxygenation; H/D: hemodialysis; HTx: heart transplantation; IABP: intra-aortic balloon pumping; ICMP: ischemic cardiomyopathy; MCS: mechanical circulatory support; SD: standard deviation; VAD: ventricular assist device.

**Table 4 jcm-10-04697-t004:** Index procedure and follow-up data in MCS bridge to HTx subgroups.

	IABP Group 2	ECMO Group 3	ECMO-VAD Group 4	VAD Group 5	*p*	Total
Patients, *n* (%)	15 (11)	33 (24.3)	59 (43.4)	29 (21.3)		136
Requiring H/D after HTx, *n* (%)	6 (40.0)	22 (66.7)	26 (45.6)	11 (37.9)	0.099	65 (48.5)
Desensitization in HTx, *n* (%)	1 (6.7)	0 (0.0)	13 (22)	4 (13.8)	0.022	18 (13.2)
Ischemic time, min (±SD)	162.5 ± 67.1	165.6 ± 58.5	175.5 ± 71.7	191.2 ± 73.6	0.439	175.0 ± 68.6
Bypass duration, min (±SD)	144.0 ± 57.4	157.1 ± 75.7	183.5 ± 60.0	183.1 ± 78.6	0.093	172.6 ± 68.9
Procedure duration, min (±SD)	336.5 ± 138.8	357.1 ± 120.0	357.7 ± 116.1	313.9 ± 98.0	0.366	345.7 ±116.2
MCS after HTx, *n*(%)	8 (53.3)	15 (45.5)	22 (37.3)	5 (17.2)	0.055	50 (36.8)
Extubation after HTx, day (±SD)	3.3 ± 5.3	9.1 ± 8.5	9.3 ± 12.2	4.6 ± 5.3	0.266	7.8 ± 9.9
ICU stay after HTx, day (±SD)	8.8 ± 3.6	18.4 ± 12.9	22.2 ± 18.8	18.1 ± 12.1	0.081	18.9 ± 15.5
Hospital survival, *n* (%)	10 (66.7)	21 (63.6)	50 (84.7)	25 (86.2)	0.051	106 (77.9)
90 days survival rate	80.0%	72.7%	88.1%	89.3%	0.093	83.7%
1-year survival rate	65.5%	57.6%	82.3%	84.6%	0.012	74.6%
3-year survival rate	57.3%	48.1%	75.1%	74.6%	0.031	65.9%
Rejection ≥ 1B in 3 years, *n* (%)	0 (0.0)	3 (12.5)	8 (14.0)	1(3.6)	0.263	12 (9.8)
AMR in 3 years, *n* (%)	0 (0.0)	1 (4.2)	3 (5.3)	1(3.6)	0.856	5 (4.1)
Rejection ≥ 1B in 5 years, *n* (%)	0 (0.0)	4 (16.7)	8 (14.0)	1 (3.6)	0.202	13 (10.7)
AMR in 5 years, *n* (%)	0 (0.0)	1 (4.2)	4 (7.0)	1 (3.6)	0.718	6 (4.9)

AMR: antibody-mediated rejection; H/D: hemodialysis; HTx: heart transplantation; IABP: intra-aortic balloon pumping; MCS: mechanical circulatory support; SD: standard deviation.

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
