# Peer review of "Impact of the Pre-Transplant Circulatory Supportive Strategy on Post-Transplant Outcome: Double Bridge May Work"

_jcm, 2021, doi:10.3390/jcm10204697_

Round 1

Reviewer 1 Report

Chou et al retrospectively reviewed 251 cases of heart transplantation in which mechanical circulatory support (MCS) was used.

Major drawbacks

  • The methodology of this work is not adequate. The cornerstone of the work is the comparison between five groups characterized by five different therapeutic strategies, but you can’t compare such different populations with different clinical conditions. The authors clearly state that “The selection of MCS devices was largely depended on the hemodynamic condition and reserve of heart function”, so the clinical outcome depends obviously on baseline conditions and only marginally on the selected therapy.

Minor drawbacks

  • Methods section is too long and can be shortened e.g. immunosuppressant protocol is irrelevant to this study

Author Response

First of all, we thank you for your kindly comment. 

1.

  Regarding this topic, we have tried every effort to delineate the effect of mechanical circulatory support (MCS) on the impact of heart transplantation on the long-term outcome. We carefully performed multivariate analysis to evaluate the factors in Tables 3 and 4, and the only risk factors were preTransplant CPR and the duration of the MCS (the data was shown in the new revised edition). 

  We realized that the process of the selection of the patients, the timing, and the types of MCS was always a bias in the process of the decision.  This is the reason why we demonstrated the past evolution of the different types of MCS in figure 1. It was an evolution in the past 12 years. It also could be debated that the concept change was the key issue for the outcome. 

  In the clinical setting, we will find any MCS to offer the adequate support as possible, and if the support is inadequate, the further advanced MCS will be applied as soon as possible. We tried to demonstrate the principle of adequate support for the end organs' function to achieve the criteria of HTx. 

  The purpose of the present study was to advocate that ECMO may consider first when patients are under emergency or CPR, and then converted to  VAD. The strategy may work as a screening protocol to optimize the patients in  a better condition to receive transplantation. 

2. 

We have deleted the immunosuppressant part in the methodology. 

Reviewer 2 Report

Extracorporeal membrane oxygenation (ECMO), one of the temporary mechanical circulatory supports (MCS)s, is an important device for resuscitation of cardiogenic shock. Temporary MCS such as Impella 5.0, Centrimag® and ECMO are relatively favored as a bridge to heart transplantation for heart failure patients with acute decompensation, categorized as INTERMACs profile 1 (Chou et al., 2006; Seese et al., 2020). Few data supported the implantation of durable ventricular assisted devices (VAD) for INTERMACs profile 1 cases considering high risks of bleeding, multiple organ failure, and surgical mortality (Pawale et al., 2018). Registration data from the international society of heart and lung transplantation showed inferior survival after heart transplantation among those who received  ECMO bridging than the survival among those who received bridging with ventricular assisted devices (VAD) (Khush et al., 2019). In this regard, double and early bridging from ECMO to VAD has been proposed since late 1990s to improve patients survival after heart transplantation (Bowen et al., 2001; Chen et al., 2021; Chou et al., 2006; Chung et al., 2010; Pagani et al., 1999).

The study was once presented as a conference abstract in the international society of heart and lung transplantation in Montreal 2020. The investigation was performed through retrospective data review from Jan 2009 to Jan 2019 in a tertiary medical center.  A toral of 251 heart transplantation patients were included and categorized in to 5 groups based on the MCS used before transplantation: non MCS (group 1), IABP (group 2), ECMO (group 3), ECMO-VAD (double bridge, group 4), and VAD (group 5). Higher prevalence of cardiopulmonary resuscitation history among ECMO than that among non-ECMO groups were found. Comparable (hospital, and 3-year) survival rates were found among group 1, 4, and 5 patients. The average duration of bridge from ECMO to VAD is 12 days followed by 51 days from VAD to transplantation. The average duration of bridge to heart transplantation is 14 days for group 3 and 128 days for group 5. The authors concluded that ECMO to VAD double bridge strategy may provide comparable post-transplantation outcome to that of single bridge strategy with VAD. The study, though not novel, may provide additional evidence to support the role of ECMO in the double bridge strategy to heart transplantation. Nevertheless, differences in the waitlist duration from initiation of MCS to heart transplantation presented as a major bias in this study. Other points that need to be addressed are as follows:

Major--

Introduction: .

  1. A brief introduction of the double bridge strategy to heart transplantation would be helpful for readers to understand the study.

Materials and Methods:

  1. Page 8, lines 1-2: could the authors specify the criteria (study definition) of "high dose" inotropes; and "adequate hemodynamics?

Results:

  1. Page 11, line 3: Does group 5 patients include both tVAD and dVAD? How about the VAD types (temporary or durable) in group 4?
  2. Page 12, lines 1-2: The authors should explain in their study the reason for better hospital survival among VAD group as compared to non-VAD group.
  3. Page 12, line 2: for hospital survival with p=0.005, the data does not match the p value (of hospital survival: 0.051) in Table 4. Please explain.
  4. May the authors comment about the incidence of cardiac allograft vasculopathy among these 5 groups of patients at follow up.
  5. Page 13, lines 13-15: What types of VAD (t or d) were used for those 10 CPR cases in VAD group? Are there double bridge (tVAD to dVAD) patients in this group?

Discussion:

  1. Page 13, lines 15-16: the authors assume that preoperative stabilization with ECMO explains for similar post-transplantation survival between group 4 and 5. Is it possible that application with VAD negatively impacted the survival outcome among VAD group, especially for those 10 CPR cases?
  2. For multivariate analysis of hospital mortality, does the presence of ECMO increase the risk ratio?
  3. The authors might provide a discussion about worse survival outcome in IABP group as that in ECMO patients using single bridge strategy to transplantation.

Author Response

We thank you for your comment and add a lot of important references for us. We already add some new references to the manuscript. 

Major--

Introduction: .

  1. A brief introduction of the double bridge strategy to heart transplantation would be helpful for readers to understand the study.

Ans:  We already add s brief introduction

The double bridge strategy is increasingly adopted. For IMTERMACS profile 1 and CS patients, initial ECMO therapy with crossover to VAD is associated with improved outcomes10-12. The double bridge strategy in heart transplantation can provide a window or screening tool to select a more suitable condition of recipients to receive a donor heart in a short -term outcome that might be helpful for the transplant center program result. In the present study, we wonder about the longer outcome and survival regarding the double bridge strategy and different MCS statuses. 

Materials and Methods:

  1. Page 8, lines 1-2: could the authors specify the criteria (study definition) of "high dose" inotropes; and "adequate hemodynamics?

Ans: we have briefly added the criteria of "high dose", the timing of MCS, and the possible options.

 IABP was reserved for patients with a mild hemodynamic disturbance under a high dose of inotropes. ECMO or VAD was implanted if the circulatory support of IABP failed to achieve an adequate hemodynamic status. In general, we applied inotropic equivalent (IE, µg/kg/min. = dopamine + doubutamine + epinephrine x 100 + norepinephrine x 100) as parameter, when IE over 15 and the systolic pressure < 90 mmHg, IABP was considered. When persistent end-organ dysfunction after IABP or IE > 20, further advanced MCS were considered. 

Results:

  1. Page 11, line 3: Does group 5 patients include both tVAD and dVAD? How about the VAD types (temporary or durable) in group 4

Ans: Yes, we add the detail number in the different groups.

Based on the types of MCS, the patients were divided into five groups: Group 1 (no MCS): 115 patients; Group 2 (IABP only): 15 patients; Group 3 (ECMO): 33 patients; Group 4 (ECMO-VAD): 59 patients, including tVAD 54, dVAD 5; Group 5 (VAD): 29 patients, including tVAD 23, and dVAD 5. With advances in MCS and the increased availability of machines, our preference for MCS has changed from IABP and ECMO to VAD and the double-bridge strategy in recent years (Fig. 1). 

  1. Page 12, lines 1-2: The authors should explain in their study the reason for better hospital survival among VAD group as compared to non-VAD group.
  2. Page 12, line 2: for hospital survival with p=0.005, the data does not match the p value (of hospital survival: 0.051) in Table 4. Please explain.

Ans: thank you for allowing me to answer these two issues together. 

First, The reason why the VAD recipients had better hospital survival than those non-VAD patients with MCS support.  we must declare that the comparison was based on those supported by MCS, not group 1 (without MCS).  In the MCS support recipients, VAD offered a more adequate circulatory support than IABP and without the possible complication of increased LV afterload. 

However, when we focused on the between patients in the VAD-supported (group 4 and group 5, total 88) group and non-VAD supported groups (group 2 and group 3, total 48), the VAD-supported group had significantly better hospital survival than the non-VAD group (p = 0.005). The analysis reconfirmed that the recipient supported with VAD had a better short-term outcome than those supported with IABP and ECMO only, although the basic demographic data had differences in the factors of previous cardiac surgery and CPR (Table 3).

About the issue of "for hospital survival with p=0.005, the data does not match the p value (of hospital survival: 0.051) in Table 4", 

We thank you for highlighting the important issue.  We are interested the long-term outcome in different types of MCS. We also fully understand the selection bias was the key issue in the process. We tried to grouping the patients with VAD (group 4 and 5) and those without VAD (group  2 and 3) and it showed the recipients with VAD had significant hospital survival.

But if we divided into 4 groups (group 2 to 5), the statistical difference is non-significant. We consider it was the grouping effect. The case number in individual groups was limited and the comparison among them did not reach difference.  In contrast, it demonstrated the difference when we regrouping the MCS recipient.  We did not use this principle in the writing because it was not reasonable in the clinical setting. We only briefly described our findings and interest in the manuscript, and hope to have some additional for the reader. 

we described in manuscript in summary because of limation of words.  

  1. May the authors comment about the incidence of cardiac allograft vasculopathy among these 5 groups of patients at follow up.

ANs: 

The incidence of allograft vasculopathy did  demostrated sginifcatly difference among the groups , Please refer to Table 4.  

  1. Page 13, lines 13-15: What types of VAD (t or d) were used for those 10 CPR cases in VAD group? Are there double bridge (tVAD to dVAD) patients in this group?

Ans:  we added detail explanation in discussion section as followed. 

The CPR cases were more challenging in those may be considered as a heart recipient. In general, ECMO was initialed at the episode of CPR, then converted to VAD or directly wait for heart when neurological status was acceptable after further assessment. There were 10 patients in group 5 (VAD only) experienced CPR but they fortunately returned of spontaneous circulation without requiring ECMO rescue, therefore they could be implanted VAD emergently. However, due to the economic issue, not everyone can afford to apply dVAD (tVAD 23, and dVAD 5). In addition, we did not have the patients bridging from tVAD to dVAD because the only tVAD was reimbursed by the insurance.

Discussion:

  1. Page 13, lines 15-16: the authors assume that preoperative stabilization with ECMO explains for similar post-transplantation survival between group 4 and 5. Is it possible that application with VAD negatively impacted the survival outcome among VAD group, especially for those 10 CPR cases?

ANs: 

Multivariate analysis was formed according to the pretransplant factors, the duration between MCS to HTx (days) and the pre-transplant CPR were the significant risk factor for hospital survival (p = 0.04 and 0.003, individually). The MCS strategy did not reach statistical difference (p = 0.058). Therefore, we believed that it was the duration of MCS leading to difference in hospital survival, and ECMO was the short-term device for MCS. 

for multivariate analysis of hospital mortality, does the presence of ECNO increase the risk ratio??

Ans: 

According to our univariate analysis in the Tables, ECMO did not show the risk ratio and the further multivariate analysis demonstrated only duration and the preCPR were the risk factor. You may argue that ECMO is the shor-term MCS, and could ECMO be a contributing factor?

Yes, it may be, but since we had group 4 (double bridge group) that may be another confounding effect. Anyway, the multivariate statistical result showed duration and preCPR were the risk. 

we add on the discussion section

 Multivariate analysis was formed according to the pretransplant factors, the duration between MCS to HTx (days) and the pre-transplant CPR were the significant risk factor for hospital survival (p = 0.04 and 0.003, individually). The MCS strategy did not reach statistical difference (p = 0.058). Therefore, we believed that it was the duration of MCS leading to difference in hospital survival, and ECMO was the short-term device for MCS. 

  1. The authors might provide a discussion about worse survival outcome in IABP group as that in ECMO patients using single bridge strategy to transplantation

Ans:

The worse survival outcome in IABP and ECMO is related to the less support than VAD did. We described briefly in the end of the discussion.  

Finally, in the clinical practice, we will find any MCS to offer the adequate support as possible, and if the support is inadequate, the further advanced MCS will be applied as soon as possible. We tried to demonstrate the principle of adequate support for the end organs function to achieve the criteria of HTx. 

Round 2

Reviewer 1 Report

The authors clearly state that "the process of the selection of the patients, the timing, and the types of MCS was always a bias in the process of the decision." The methodology bias previously reported has not been completely resolved. 

Author Response

The authors clearly state that "the process of the selection of the patients, the timing, and the types of MCS was always a bias in the process of the decision." The methodology bias previously reported has not been completely resolved. 

Ans:

Thank you for your critical comment. We tried to explain the general principle in detail we usually applied in the process of the decision, selection of the types of MCS in the methodology. As I described, this is a general principle that may be changed when the different patients' conditions. 

we also add this point in the limitation to clarify the possible drawback of the manuscript. 

The selection of the types of MCS was a process of decision-making, including the status of numbers of organs dysfunction, status of the INTEERMAC, the availability of the MCS, the supporting capacity of the different MCS, and the predicted duration of the waiting time. Although there were lots of uncertain factors remaining to consider, we tried to describe the general rule that we usually applied.

    IABP was the firstly MCS that we took into consideration. However, because of the limited additional supportive capacity, it was reserved for patients with a mild hemodynamic disturbance under a high dose of inotropes. After that, ECMO or VAD were implanted if the circulatory support of IABP failed to achieve an adequate hemodynamic status. In general, we applied inotropic equivalent (IE, µg/kg/min. = dopamine + doubutamine + epinephrine x 100 + norepinephrine x 100) as parameter, when IE over 15 and the systolic pressure < 90 mmHg, IABP was considered. When persistent end-organ dysfunction after IABP or IE > 20, further advanced MCS were considered. 

    For patients of INTERMACS profile 1 with CS or under CPR, ECMO was set up as the resuscitation of choice. The ECMO equipment, management, and exclusion criteria have been described in previous studies13. If a lung edema or marked LV distension developed without aortic valve opening despite optimal medication after ECMO, a LVAD was required. In this double-bridge strategy, we usually choose a tVAD (Levitronix CentriMag®) as a bridge to recovery, decision or transplantation. For patients without signs of heart function recovery and for whom heart transplantation was inevitable, we also shifted to VAD after stabilization of renal and liver function. 

    For patients of INTERMACS profile 2 or 3 who were inotrope-dependent, we applied VAD as a bridge to heart transplantation. The VAD systems that we adopted included temporary VAD (tVAD): CentriMag™ (Abbott, Santa Clara, Calif) and durable VAD (dVAD): HeartMate II™, HeartMate 3™ (Abbott Laboratories, Lake Bluff, Ill) and HeartWare™ (Medtronic, Mounds View, MN). We usually placed the outflow cannulae at the ascending aorta or subclavian artery, with inflow from the left ventricular apex for tVAD. The implanting procedure for dVAD was performed as recommended14.

Limitation
    This study is a single-center, retrospective record review with limited case numbers, and the results must be interpreted with caution. In addition, the process of the selection of the patients, the timing, and the types of MCS was always a bias in the process of the decision.

Reviewer 2 Report

The authors had provided point-to-point responses. No further comments.

Author Response

Dear reviewer 2:

Thank you for your kindness to accept our viewpoint and manuscript.